# Recapitulation of the accessible interface of biopsy-derived canine intestinal organoids to study epithelial-luminal interactions

Yoko M. Ambrosini[1,2][☉], Yejin Park[3][☉], Albert E. Jergens[4], Woojung Shin[1], Soyoun Min[1], Todd Atherly[4], Dana C. Borcherding[2], Jinah Jang[3,5,6], Karin Allenspach[4], Jonathan P. Mochel[2]*, Hyun Jung Kim[1,7]*

1 Department of Biomedical Engineering, The University of Texas at Austin, Austin, TX, United States of America, 2 Department of Biomedical Sciences, Iowa State University, Ames, IA, United States of America, 3 Department of Creative IT Engineering, Pohang University of Science and Technology, Pohang, Korea, 4 Department of Veterinary Clinical Sciences, Iowa State University, Ames, IA, United States of America, 5 School of Interdisciplinary Bioscience and Bioengineering, Pohang University of Science and Technology, Pohang, Korea, 6 Department of Mechanical Engineering, Pohang University of Science and Technology, Pohang, Korea, 7 Department of Oncology, Dell Medical School, The University of Texas at Austin, Austin, TX, United States of America

☉ These authors contributed equally to this work.
* hyunjung.kim@utexas.edu (HJK); jmochel@iastate.edu (JPM)

**Data Availability Statement:** All relevant data are within the paper and its Supporting Information files.

## Abstract

Recent advances in canine intestinal organoids have expanded the option for building a better *in vitro* model to investigate translational science of intestinal physiology and pathology between humans and animals. However, the three-dimensional geometry and the enclosed lumen of canine intestinal organoids considerably hinder the access to the apical side of epithelium for investigating the nutrient and drug absorption, host-microbiome crosstalk, and pharmaceutical toxicity testing. Thus, the creation of a polarized epithelial interface accessible from apical or basolateral side is critical. Here, we demonstrated the generation of an intestinal epithelial monolayer using canine biopsy-derived colonic organoids (colonoids). We optimized the culture condition to form an intact monolayer of the canine colonic epithelium on a nanoporous membrane insert using the canine colonoids over 14 days. Transmission and scanning electron microscopy revealed a physiological brush border interface covered by the microvilli with glycocalyx, as well as the presence of mucin granules, tight junctions, and desmosomes. The population of stem cells as well as differentiated lineage-dependent epithelial cells were verified by immunofluorescence staining and RNA *in situ* hybridization. The polarized expression of P-glycoprotein efflux pump was confirmed at the apical membrane. Also, the epithelial monolayer formed tight- and adherence-junctional barrier within 4 days, where the transepithelial electrical resistance and apparent permeability were inversely correlated. Hence, we verified the stable creation, maintenance, differentiation, and physiological function of a canine intestinal epithelial barrier, which can be useful for pharmaceutical and biomedical researches.

**Funding:** This work was supported in part by the Burroughs Wellcome Fund Collaborative Research Travel Grant (BWF 1019990.01 to Y.M.A.), the American College of Veterinary Internal Medicine Advance Research Fellowships (K.A. and Y.M.A.), the Ministry of Science and ICT Korea under the ICT Consilience Creative program supervised by the Institute for Information & communications Technology Planning & Evaluation (IITP) (IITP-2019-2011-1-00783 to Y.P. and J.J.), Basic Science Research Program through the National Research Foundation of Korea funded by the Ministry of Education (2015R1A6A3A04059015 to Y.P. and J.J.), Iowa State University (Faculty startups, College of Veterinary Medicine Seed Grant and Miller Award), Bio & Medical Technology Development Program of the National Research Foundation funded by the Ministry of Science and ICT (2018M3A9H3025030 to H.J.K.), Technology Impact Award of the Cancer Research Institute (UTA18-000889 to H.J.K.), the National Cancer Institute of the National Institutes of Health, IMAT program (R21CA236690 to H.J.K.), the Leona M. & Harry B. Helmsley Charitable Trust (Grant #1912-03604 to H.J.K.), and F99/K00 Predoctoral to Postdoctoral Transition Award (F99CA245801 to W.S.), and Asan Foundation Biomedical Science Scholarship (W.S.). 3D Health Solutions provided support in the form of equity interest for J.P.M., A. E.J., K.A., and H.J.K and in the form of a salary for T.A. The funders had no role in study design, data collection and analysis, decision to publish, or preparation of the manuscript. The specific roles of these authors are articulated in the 'author contributions' section.

**Competing interests:** J.P.M., A.E.J., K.A., and H.J. K. are co-founders of 3D Health Solutions Inc. and hold an equity interest in the company. T.A. is employed by and receives a salary from 3D Health Solutions Inc. This does not alter our adherence to PLOS ONE policies on sharing data and materials.

# Introduction

Multiple chronic human disorders, including inflammatory bowel disease (IBD) and colorectal cancer (CRC), have been characterized in canine models based upon the spontaneous clinical analogs of gastrointestinal (GI) disorders [1,2]. For the investigation of human intestinal homeostasis, canine models are especially relevant to humans because their intestinal physiology and diet style have adapted to those of humans during domestication [3]. Due to this similarity, it is not surprising that dogs and humans share similar composition of the gut microbiota with ~60% taxonomic and functional overlap as compared to <20% for mice [4]. Therefore, dogs are considered a more predictable animal model for investigating environmental influences on human GI health and disease compared to conventional murine models [4].

There is currently a limited number of canine-specific primary cell lines to investigate intestinal physiology *ex vivo* or *in vitro*. Well-characterized immortalized cell lines including the Madin-Darby canine kidney (MDCK) cells do not accurately model intestinal epithelial interactions in the dog due to their origin from immature kidney cells [5,6]. Recently, isolated primary canine intestinal epithelial cells have been immortalized with a temperature-sensitive mutant of the Simian Virus 40 large tumor antigen (SV40 T-Ag) [7]. Although this cell line can be grown on a monolayer, the SV40 T-Ag may initiate pathways which could provide spurious, non-physiologic findings *ex vivo* given its tumorigenic cell line origin [8].

We have recently optimized the three-dimensional (3D) *in vitro* culture conditions of canine primary intestinal organoids and shown that isolated intestinal stem cells differentiate into organoids containing matured intestinal cell lineages within ~8 days of culture [9]. The 3D organoid culture technology not only offers a more physiological platform compared with conventional 2D cell lines [10], but also provides a "personalized" modeling to investigate the effect of environmental stimuli or dietary interventions on intestinal epithelium [11]. Altogether, the establishment of a robust canine organoid protocol allows for comparative biomedical initiatives in humans and dogs to be performed [2].

However, a notable limitation of the 3D intestinal organoid system has been identified. For instance, the 3D organoid body prevents the access to the lumen for studying the interactions with dietary constituents, microorganisms, drugs, or toxins transported through an epithelial layer [12]. While microinjection of a luminal component (e.g., living bacterial cells) into the lumen of an organoid has been feasible, the technique can be challenging due to the heterogeneity in organoid size, invasive injection, and the requirement of techniques and equipment [13]. Thus, cultures of a polarized intestinal cell monolayer are better suited for the standardized measurement of transepithelial permeability and epithelial-luminal interaction due to easier accessibility of the apical surface. Moreover, creating a canine-derived intestinal interface may be further improved by integrating the optimized protocol to the intestinal microphysiological systems [14–17].

In this study, we report an optimized method for generating an intact monolayer of the canine colonoid-derived epithelium. We characterized the formed epithelial monolayer that provides an accessible tissue interface, polarization, lineage-dependent differentiation, tight junction barrier, permeability, and the expression of key efflux pump using various imaging modalities. We envision that our optimized protocol and the robust culture of canine-derived epithelium may enable to develop an advanced *in vitro* model to demonstrate complex host-gut microbiome crosstalk and pharmacological assessment under various disease milieus.

# Results

## Recreating a canine colonoid-derived intestinal tissue interface

Canine colonic organoids derived from three independent canine donors were expanded in 3D geometry for up seven days in Matrigel (Fig 1A), allowing a long-term culture and storage

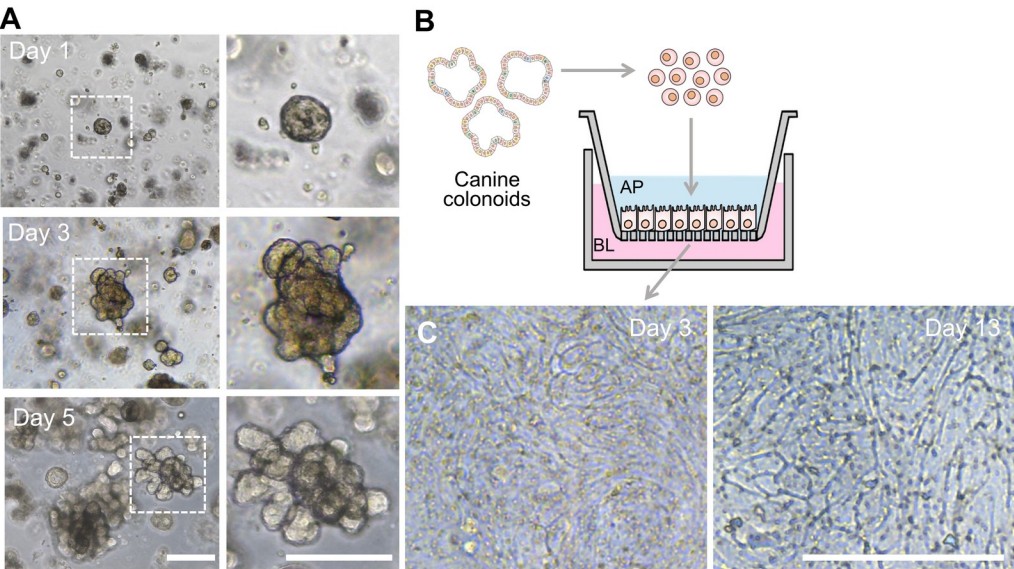

**Fig 1. Morphological analysis of the 3D colonoids and the 2D canine colonic monolayer.** (**A**) A growth profile of the colonoid isolated from the canine colonic crypt. A small spherical colonoids progressively grows to form fully grown colonoids. Representative phase-contrast micrographs were taken at days 1, 3, and 5. The zoomed-in inset at each day shows the high-power magnification of a colonoid in the white dashed box. (**B**) A schematic displays the procedure of the formation of an epithelial monolayer derived from 3D canine colonoids. The fully-grown organoids are dissociated into single cells, then seeded into a nanoporous insert to form a monolayer. AP, apical; BL, basolateral. (**C**) Representative phase-contrast micrographs on day 3 and 13 are provided, respectively. Bars, 200 μm.

of the primary intestinal epithelium [9]. A colonoid-derived monolayer was generated in a nanoporous insert of the Transwell pre-coated with the extracellular matrix (ECM) mix with Matrigel (100 μg/mL) and collagen I (30 μg/mL) by introducing the dissociated colonoid cells (Fig 1B). In terms of the colonoid dissociation, we employed an enzymatic dissociation method [15] to generate single-cell suspension to accomplish a confluent monolayer, which can be maintained for at least 13 days (Fig 1C).

## Apical microvilli formation in the canine colonic epithelial monolayer

The polarization of the colonic epithelium is critical to establish a biological tissue interface. Microvilli that illustrate the polarized apical membrane of the colonic epithelium were observed on the recreated monolayer using scanning electron microscopy (SEM; Fig 2A and 2B) and transmission electron microscopy (TEM; Fig 2C and 2D). A variation in microvilli frequency was observed in our dog colonoid-derived monolayer, which was also noted in other colonoid-derived studies [18,19]. The number of microvilli assessed by the SEM imaging was variable in the range from 9 to 18 microvilli/$\mu m^2$, which was similar to the reports of human intestinal epithelial cell culture performed *in vitro* [20]. Glycocalyx, which provides a physical glycosylated barrier on the epithelial cells [21], was also well generated at the surface of the microvilli (Fig 2D).

## Lineage-dependent characterization of the differentiated canine colonic epithelial monolayer

RNA *in situ* hybridization (RNA-ISH), immunofluorescence (IF), and electron microscopic imaging were used to show the differentiated cell lineages in the canine colonoid-derived monolayer. The leucine-rich repeat-containing G-protein coupled receptor 5 (*Lgr5*), a seminal

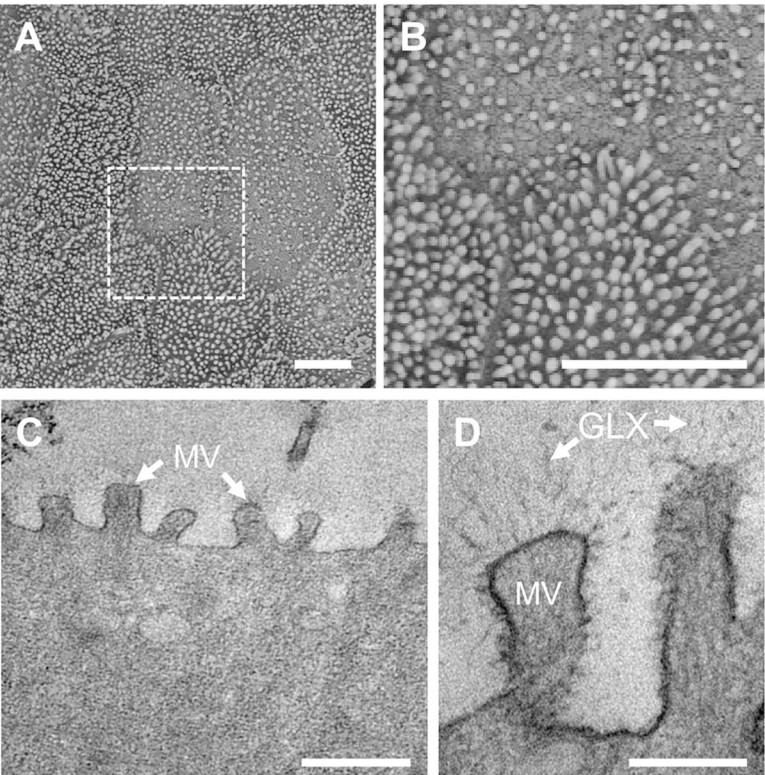

**Fig 2. Electron microscopic characterization of the apical surface and the tissue interface of the canine colonoid-derived monolayer.** (**A**) A low magnification SEM image of the microvilli on the apical cell surface. (**B**) A high-power magnification of the microvilli from **A** indicated by a white dashed box. Bars, 5 μm. (**C**) A TEM image of the microvilli on the cell monolayer. MV, microvilli. Bar, 500 nm. (**D**) A high-power TEM image that shows the microvilli (MV) and the surrounding glycocalyx (GLX). Bar, 200 nm.

marker for adult intestinal stem cells [22], was detected sporadically in the 2D monolayer cultured for 14 days (Fig 3A). Also, the canine colonic epithelium retained a population of proliferative cells, as visualized by Ki67-positive signals for up to 2 weeks (Fig 3B). The differentiated absorptive enterocytes were visualized by the staining with intestinal alkaline phosphatase (*ALPI*) (Fig 3C) [23]. The enteroendocrine cells were highlighted using Neurogenin 3 (*Neurog3*; Fig 3D) and Chromogranin A markers (CgA; Fig 3E), respectively [24]. In the canine epithelial monolayer, we analyzed the appearance of each cell type based on the imaging results, where the *Lgr5* stem cells, Ki67+ proliferating cells, *ALPI*+ differentiated intestinal epithelium, *Neurog3*+ and CgA+ enteroendocrine cells were populated as 7.6±0.1, 38.4±2.4, 60.1±0.9, 41.2±10.3%, and 47.8±2.7%, respectively (Fig 3F).

To investigate the presence of physiological mucus production in the monolayer, live-cell staining with Wheat Germ Agglutinin (WGA) was performed [17,25,26]. We found that the WGA-positive signals were detected across the entire monolayer, suggesting that the epithelial apical surface was covered by mucus-like molecules such as *N*-acetyl-D-glucosamine (Fig 4A). We also identified the mucin granule-containing goblet cells using TEM (Fig 4B, "MG"), where the goblet cell orifices (Fig 4D, "GO") and fenestrated membranes (Fig 4D, "FM") extending deep into the goblet cell were also confirmed using SEM, as shown in previous studies [27,28], demonstrating that the goblet cells were present in the canine colonoid-derived monolayer.

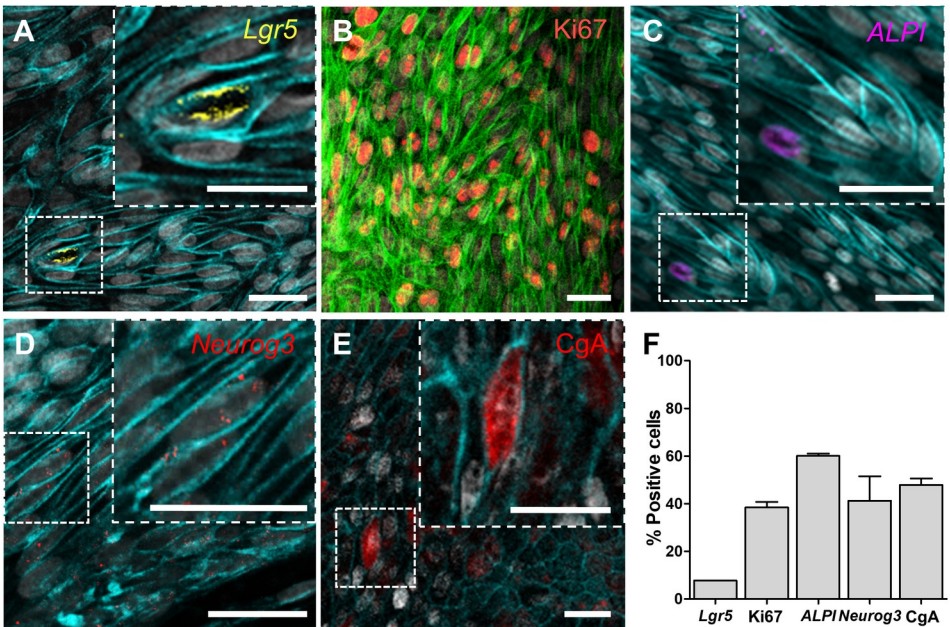

**Fig 3. The cell type-specific characterization of the canine colonoid-derived epithelial monolayer.** The canine colonoid-derived monolayer on Day 13 was used to visualize the markers highlighting the cell lineages, proliferation, and mucus production. The population of stem cells (A; *Lgr5+*, Yellow), proliferative cells (B; Ki67, Red), absorptive enterocytes (C; *ALPI*, Magenta), and enteroendocrine cells (D; *Neurog3*, Red and E; CgA, Red) were visualized by using RNA *in situ* hybridization (for A, C, and D) or IF staining (for B and E). As a counterstaining, E-cadherin (Cyan for A, C, and D), F-actin (Green for B and Cyan for E), or nuclei (Grey for A, B, C, D, and E) were displayed. Bars, 20 μm. (F) Quantification of the population of the cells that show positive signals to the target markers normalized by the total numbers of nuclei. Three independent fields of view from two or more independent biological replicates were used. In each biological replicate, 2 technical replicates were performed. Error bars indicate SEM.

In addition, we confirmed that the P-glycoprotein (P-gp) efflux transporters were diffusely expressed on the apical surface of the canine colonoid-derived monolayer (Fig 5A and 5B), which is consistent with the localization of the P-gp transporters in the canine colonic tissue [29]. Importantly, the IF assessment revealed that the polarized expression of P-gp was significantly ($P < 0.0001$) increased on Day 13 compared to the images acquired on Day 3 on the nanoporous insert, suggesting that the maturity of the colonoid-derived epithelial monolayer was achieved (Fig 5C).

## Assessment of the canine intestinal barrier integrity

The formation of tight-junction proteins was confirmed by IF staining for zonula occludens 1 (ZO-1) (Fig 6A) and E-cadherin (E-cad) expression (Fig 6B), where no significant difference of the expression at Day 3 and 13 was observed in both ZO-1 and E-cad (S2 Fig). After 4 days of cultures, the confluent colonoid monolayer showed stable transepithelial electrical resistance (TEER) values of approximately 1,000 Ω·cm² (Fig 6C). We observed that the TEER value was stably maintained for up to 14 days when the culture medium was replenished every other day for all 3 independent lines of canine colonoid-derived epithelium (S3 Fig).

Next, we evaluated the effect of the complete medium with or without Wnt proteins on the growth of canine colonoid-derived monolayer to verify the effect of differentiated culture condition on the epithelial barrier function. Briefly, the overall profile of TEER cultured in both the differentiation (i.e., the Wnt-free and Wnt-containing medium in the apical and basolateral compartment, respectively; Fig 6C, "Diff") and proliferation medium (i.e., Wnt-

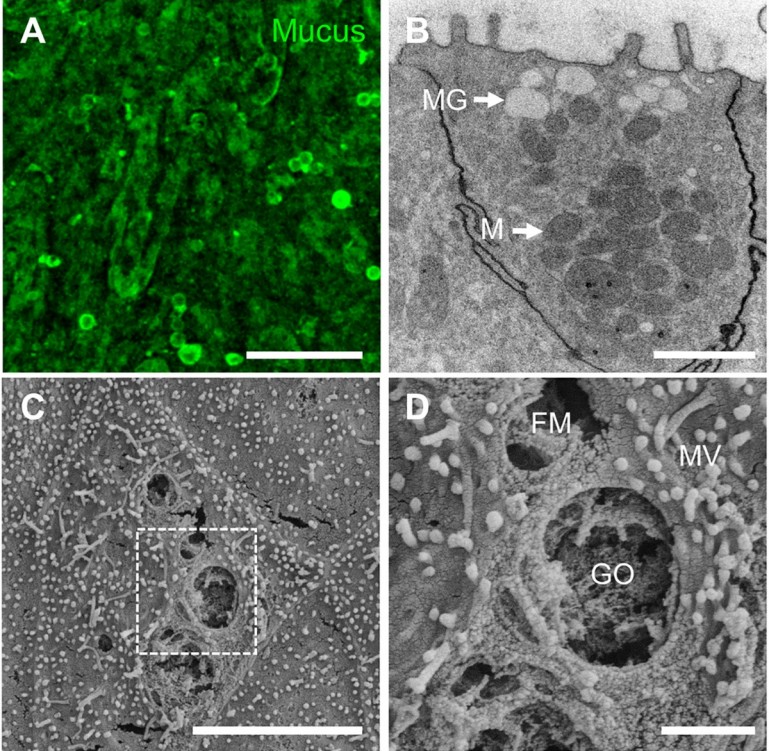

**Fig 4. Visualization of mucus production and goblet cells in the canine colonoid-derived monolayer.** The mucus production (**A**; WGA) was visualized by live-cell imaging at the apical surface of the monolayer. Bar, 20 μm. (**B**) A representative TEM image shows the goblet cell with multiple mucin granules (MG) and mitochondria (M). Bar, 1 μm. (**C**) A low magnification SEM image of a goblet cell on the apical cell surface of the canine colonoid-derived monolayer. Bar, 5 μm. (**D**) A high magnification of a goblet cell orifice (GO), a fenestrated membrane (FM) extending deep into the cell, and microvilli (MV) from **C** indicated by a white dashed box. Bar, 1 μm.

containing medium to both compartments; Fig 6C, "Control") showed a similar decline as a function of time. However, the monolayer conditioned under the differentiation medium showed a temporal maintenance of the TEER for ~2 days compared to the control ($P < 0.01$). The effect of different culture medium on the TEER values became negligible over time by

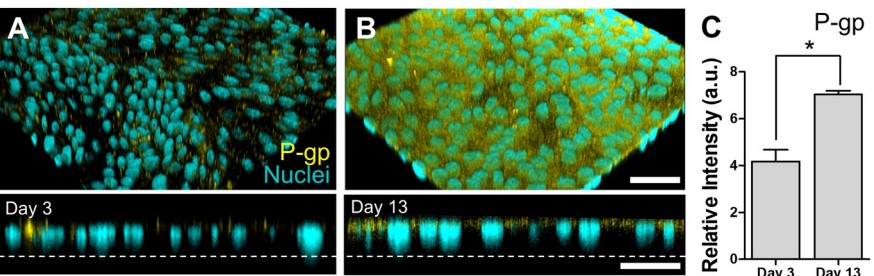

**Fig 5. Expression of the P-gp in the canine colonoid-derived monolayer.** The expression of P-gp was visually characterized by IF staining. Angled (upper) and cross-sectional side views (lower) show the localization the P-gp proteins (Yellow) on the polarized colonoid-derived monolayer at days 3 (**A**) and 13 (**B**), respectively. Nuclei, Cyan. Dashed lines pinpoint the location of the basement membrane in the nanoporous insert. Bars, 50 μm. (**C**) Quantification of the P-gp expression at days 3 and 13, respectively. Total 10 randomly chosen fields of view were used to detect P-gp expression levels among 4 biological replicates. In each biological replicate, we performed 2 technical replicates. a.u., arbitrary unit. Error bars indicate SEM.*$P < 0.0001$.

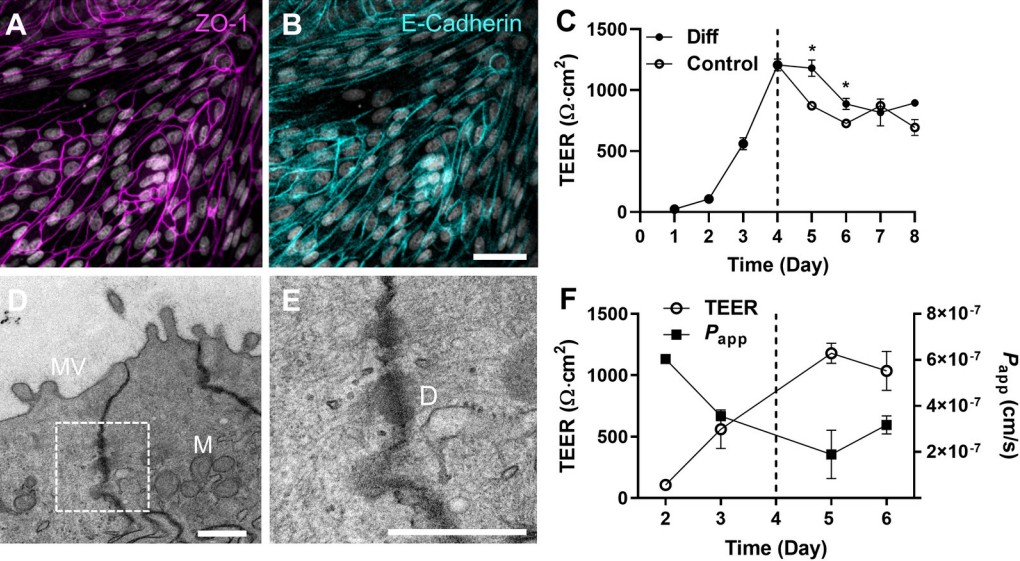

**Fig 6. Characterization of the junctional proteins and the barrier function in the canine colonoid-derived epithelial monolayer.** Visualization of the spatial localization of the ZO-1 (**A**; Magenta) and E-cadherin (**B**; Cyan) on the same location of a canine colonoid-derived monolayer. Nuclei, Grey. Bar, 50 μm. (**C**) The profile of the epithelial barrier function was monitored by measuring TEER. The effect of culture medium on TEER was demonstrated by applying the regular proliferation medium in both the apical and basolateral side of the Transwell (Control, open circle) versus the differentiation/proliferation medium in the apical/basolateral compartments, respectively (Diff; closed circle). Both groups were cultured with the proliferation medium by Day 4 (a dashed line), then different culture media were applied (Diff vs. Control) for additional 4 days. Two biological replicates with 4 technical replicates were used in each condition. $^{*}P<0.01$. (**D**) A TEM image of the intercellular junctional complex in the canine colonoid-derived monolayer and a zoom-in (**E**) that shows a high-power magnification of the white dashed area in **D**. MV, microvilli; M, mitochondria; and D, desmosome. Bars, 500 nm. (**F**) The profile of TEER (open circle) and corresponding apparent permeability ($P_{app}$) of fluorescein (closed square) on the days of 2, 3, 5, and 6 of the cultures. Each data point was prepared with 2 biological and 4 technical replicates. Error bars indicate SEM.

Day 7 (Fig 6C). This observation is consistent with the previous findings from our group where low Wnt3a-containing medium (i.e., differentiation medium) was not necessary for the development of mature canine tight junctions [9]. The TEM images revealed the presence of intercellular junctions as well as desmosomes at Day 13 (Fig 6D and 6E). Corresponding apparent paracellular permeability ($P_{app}$) to fluorescein sodium salt ($M_w$, 376.27 Da) was measured, and an inverted relationship of TEER and $P_{app}$ values was observed (Fig 6F). Specifically, as TEER values significantly increased from Day 2 to Day 6 ($P <0.0001$), corresponding $P_{app}$ values significantly decreased ($P <0.0001$), supporting that the TEER value may be used to predict the appropriate point to perform epithelial-luminal interactions.

## Discussion

In this study, we report for the first time the development of an optimized method for the generation of an intact canine colonoid-derived monolayer from canine 3D colonoids. The enzymatic dissociation method can be applied to canine organoids as performed in other species [15,30] to generate single-cell suspension to accomplish a confluent monolayer. The multimodal imaging techniques employed in this study confirmed the creation and stable maintenance of the 2D canine intestinal epithelial monolayer on a nanoporous insert up to two weeks with a physiological expression of structural tight-junctions and marker proteins.

Findings from TEM and SEM micrographs demonstrated the formation of a physiological brush border interface and the presence of glycocalyx on the microvilli, which is the

characteristic of terminally differentiated canine intestinal epithelium [31]. We confirmed that the canine epithelium cultured on a nanoporous insert grew into multiple lineages of the differentiated intestinal epithelium including absorptive enterocytes, goblet cells, and enteroendocrine cells. Furthermore, our IF imaging data confirmed that P-gp efflux proteins were apically expressed similarly to canine colonic tissue *in vivo* [29], which disseminates a follow-up study in terms of the functional assessment of P-gp efflux pumps for pharmacological applications.

We confirmed that stable TEER values could be established by Day 4 of the monolayer culture, which is similar to the previous study using canine [8] or human cell lines [32]. The TEER values increased as a concurrent decrease in the apparent permeability of a paracellular marker similar to the previous study [8], suggesting that the ideal timeline to perform the barrier-associated experiments can be estimated once stable TEER values are achieved (here, after Day 4). In human intestinal organoid culture, Wnt protein-rich medium produced largely undifferentiated progenitors due to the central role that Wnt signaling plays in the maintenance of an undifferentiated crypt progenitor state [33,34]. We demonstrated minimal effect of low Wnt3a-containing medium (i.e., differentiation medium) for the development and maintenance of mature canine tight junctions as reported previously [9].

Moreover, we showed that the canine colonoid on the Transwell contain a stable population of the intestinal stem cells as well as other differentiated cells present in the intestinal tissue of origin [9]. Using RNA-ISH imaging technology, we were able to investigate the percentage of cells expressing multi-lineage cell differentiation RNA markers, including the *Lgr5*+ stem cells [9,35], *ALPI*+ differentiated intestinal epithelium [9,28], *Neurog3*+ enteroendocrine cells [36,37], which were all similar to what have been previously reported in human and dog *in vitro* intestinal systems. It is noted that the Ki67+ cells are not the population of lineage-dependent cells; however, we included in the same chart (Fig 3F) to provide a quantitative information. It is also critical to confirm the production of intestinal mucus and a glycocalyx on the engineered epithelial monolayer [19]. We demonstrated the presence of mucus with WGA staining and the presence of glycocalyx using TEM imaging. The presence of goblet cells was also demonstrated using TEM and SEM by detecting multiple mucin granules (MG) (Fig 4B) and goblet cell orifices (GO) as well as a fenestrated membrane (FM) (Fig 4D) as shown in previous studies [27,28].

A key advantage of the recreation of a 2D mucosal tissue interface is that this culture format will allow the access to the apical side of the epithelium for investigating the nutrient and drug absorption, host-microbe crosstalk, or drug metabolism and toxicity testing. The 2D mucosal tissue interface using primary 3D intestinal organoids could allow modeling of intestinal physiology *ex vivo* or *in vitro* compared to currently available canine-specific immortalized cell lines [5–7]. The measurement of the epithelial barrier function (e.g., TEER) is convenient when investigating the physiological responses of epithelial cells following exposure to toxins, therapeutic drugs, or nutrients [38,39]. On the contrary, the conventional 3D organoid culture method in Matrigel considerably prevents the access to the apical side of the epithelium [12], which hinders the aforementioned physiological reactions such as host-microbe crosstalk. Moreover, the Matrigel for the conventional culture of 3D organoid interferes with the compound exposure to the cells, and therefore, holds limited scalability for high-throughput pharmacological applications.

The physiological multi-lineage differentiation of a colonoid-derived monolayer suggests that patient-derived organoids on a 2D Transwell platform could potentially be used for translational researches targeting Precision Medicine purposes because the individual canine organoid lines obtained from various dog species can be established and utilized for pharmaceutical studies [10]. Comparative studies using various pharmacological agents or

clinically approved drug compounds will bring translational value to bridge between experimental canine models to *in vitro* human models, then ultimately toward human *in vivo*. For instance, the Caco-2 human intestinal epithelial cell line, which has been predominantly used in the pharmaceutical industry [40], can be a good comparative model to interrogate the potential of canine organoid-derived epithelium for validating physiological and toxicological responses of the canine epithelium.

Another potential application that we envision is to leverage this proof-of-principle study of canine intestinal organoids towards the possible application of an advanced gut-on-a-chip microphysiological platform [14,41]. Over the past decade, numerous human organ- or tissue-on-a-chip models have been suggested and evaluated, whereas no significant animal-derived models have been developed. As for GI models, a couple of compelling gut-on-a-chip models have showcased that modeling physiologically relevant host-microbiome interactions is possible [16,19,41–44], as well as pathomimetic inflammatory disease modeling [16,17,45], or the co-culture of anaerobic gut bacteria in an anoxic-oxic interface [46]. Based on this newly established culture protocol, applications of the canine organoid-derived epithelial cells may help to develop a novel "Canine gut-on-a-chip" platform to demonstrate host-microbiome ecosystem and validate drug efficacy and toxicity.

One primary concern of the human biopharmaceutical industry lies in the fact that most preclinical studies fail to accurately predict the efficacy and safety of drug candidates in human clinical trials [47]. This concern is most likely due to the large translational gap between highly inbred or genetically-modified laboratory animals and humans who exhibit genetic variability influenced by environmental factors [1]. Therefore, an interdisciplinary collaboration between basic scientists, engineers, and clinician-scientists using companion animals with naturally occurring diseases is critical to bridge this gap and accelerate drug development [48]. Dogs are receiving more attention as a relevant translational *in vivo* model compared to rodents because they share similar genetic and environmental variations seen in humans [1]. For example, human intestinal disorders such as inflammatory bowel disease, ulcerative colitis, and colorectal cancer have been well characterized in clinical analogs of dogs [49–51]. Moreover, dogs and humans hold a strikingly similar composition of the gut microbiota; therefore, dogs are likely a more predictable model for studying microbial influences modulating human intestinal homeostasis [4]. The canine colonoid-derived monolayer that we report herein provides an intestinal tissue interface with multi-lineage cell differentiation, which is known to be expressed in primary tissues. This tool can also help reduce the number of dogs required to test intestinal physiology in health and disease, by the *Three Rs Principle*: Reduce, Replace, and Refine [52].

Although dogs are excellent animal models to study human diseases [1], dog studies are often limited by the number of commercially available reagents targeting major proteins shown to be relevant in mice [53,54]. The RNA-ISH technology provides an *in situ* analysis of biomarkers within the histopathological context of biological samples as they target the mRNA of select proteins [55]. RNA-ISH is a suitable alternative to IF in those cases where the detection of proteins lacks sensitivity or cellular resolution [55,56]. The customized probes for RNA-ISH can be engineered based on any RNA sequences, which allows investigators to overcome the lack of canine-specific reagents for the identification of intestinal stem cells and their lineage cells in dogs [9,55]. However, as RNA-ISH only detects mRNA expression, it provides no spatial information on actual protein expression or matured protein productive function in the cell. Regardless of the location of the positive signal, a positive signal is an indicative of the presence of the target gene(s) in that particular cell. This RNA-ISH technology has been successfully applied in dog organoids by our group [4] and similar findings (i.e., positive signals

seem to be expressed in the nucleus) can be found in other studies [9,11] as well as our positive control provided in S1 Fig.

Stunted microvilli were observed in our system which could reflect the fact that colonic intestinal cells may not require longer microvilli due to minimal nutrient absorption in the colon [57]. Possibly, it could be due to the culture condition that is not completely adequate to promote longer microvilli [28]. As described before, Wnt-enriched medium produced largely undifferentiated progenitors comprising organoids in human intestinal organoid culture [33]. Our previous [9] and current work demonstrated that canine intestinal organoids are indeed capable of differentiating into functional epithelial cells even under Wnt-enriched condition; however, the effect of low Wnt-containing medium (i.e., differentiation medium) particularly on microvilli length would be beneficial to better understand the physiological demonstration and functions of the microvilli in the future study.

Our study demonstrates the methods to create the accessible apical surface of the intestinal epithelium generated from canine colonoids. In the future study, we will investigate epithelial-luminal interactions perturbed by microbial, metabolomic, and pharmacological stimulations that mediate GI health and disease. Moreover, the method developed herein can be applied to other segments of organoids (i.e., enteroids) as well as the organoids obtained from both diseased and other healthy dogs to enable segmental investigation of epithelial-luminal interactions.

## Materials and methods

### Creation of a biopsy-derived canine colonoid line

Intestinal biopsies were obtained via colonoscopy for intestinal stem cell isolation from healthy research colony dogs at the Iowa State University College of Veterinary Medicine. All animal procedures in this study were approved by the Iowa State University Institutional Animal Care and Use Committee (IACUC protocol: 9-17-8605-K). Colonic crypts containing primary adult intestinal stem cells were isolated and cultured, as previously described [9]. Briefly, endoscopic biopsy samples from colonoscopies were cut into small pieces, and intestinal crypt cells were released by incubating the samples with a complete chelating solution and EDTA (30 mM; Alfa Aesar) at 4°C for 60 min. After the crypt release, the crypt-containing pellet was suspended and seeded in 30 μL per well of Matrigel (Corning) and 500 μL per well of complete medium supplemented with intestinal stem cell (ISC) supporting factors including 10 μM rho-associated kinase inhibitor (ROCKi) Y-27632 (StemGent) and 2.5 μM glycogen synthase kinase 3β (GSK3β) inhibitor (StemGent) before the plate was incubated at 37°C [9]. The culture medium was changed to complete medium without any supplementation after 2 days of crypt isolation.

### Colonoid culture

A basal medium containing 10 mM HEPES (Gibco), 1× GlutaMAX (Invitrogen), 100 units/mL penicillin, and 100 μg/mL streptomycin in Advanced DMEM/F12 (Gibco) was first prepared. Conditioned medium was prepared by culturing Wnt3a-producing L cells (ATCC, CRL 2647), R-spondin1 (Rspo1) cells (Trevigen), and Noggin secreting cells (Baylor's College of Medicine), as previously described [11]. In the complete medium, the volume ratio of basal and each conditioned medium is defined at 20/50/20/10% (v/v) and murine recombinant epidermal growth factor (EGF) (50 ng/mL; Peprotech), SB202190 (30 μM; Sigma Aldrich), A-8301 (500 nM; Sigma Aldrich), Gastrin (10 nM; Sigma Aldrich), *N*-acetylcysteine (1 mM; MP Biomedicals), nicotinamide (10 mM; Sigma Aldrich), N2 (1×; Gibco), and B27 (1×; Gibco) were also supplemented. The complete medium was changed every other day, and organoids

were passaged once a week by mechanically breaking down the organoids, spinning down the fragmented organoids (100× $g$, 4˚C, 5 min), resuspending centrifuged organoids with fresh Matrigel on ice, and then plating them in each well of a 24 well plate (Corning).

## Culture of a colonoid-derived monolayer

The 3D colonoids were harvested from Matrigel after 7 days of culture by addition of EDTA solution (0.5 mM; Alfa Aesar) on ice, then transferred in 15 mL tubes and centrifuged (100× $g$, 4˚C, 5 min). The organoid pellet was incubated in 1 mL TrypLE Express (Gibco) for 10 min while shaking at 37˚C in a water bath. The centrifuged (100×$g$, 4˚C, 5 min) organoid fragments were resuspended in complete medium [11] and further dissociated by repeated pipetting and subsequent filtering of the cell suspension through a cell strainer (cut-off size, 40 μm, Corning) to obtain a single-cell suspension. Transwell inserts (0.4 μm pores, Corning) were pre-coated with Matrigel (100 μg/mL; Corning) and collagen I (30 μg/mL; Fisher Scientific) in PBS or basal medium at 37˚C for 1 h. Dissociated cells were counted manually using a cell counter (Hemocytometer; Hausser Scientific) and seeded at $10^6$ cells/mL in pre-coated Transwell inserts. After 3 days of incubation in a humidified incubator at 37˚C with 5% $CO_2$, the cell monolayer was established. The morphology of a cell monolayer was intermittently monitored for up to two weeks by phase-contrast microscopy (Axiovert 40CFL, Zeiss).

## Evaluation of the epithelial barrier integrity

The barrier function of the intestinal epithelial monolayer was measured by monitoring TEER. The TEER value was measured by using Ag/AgCl electrodes connected to an Ohm meter (Millicell ERS-2; Millipore). Normalization of TEER was performed following the equation as, TEER = $(\Omega_t - \Omega_{blank}) \times A$, where $\Omega_t$ is the resistance (in Ohms) at the measured time point since the start of the culture; $\Omega_{blank}$ is the resistance of the blank, and A is the surface area cultured on the nanoporous insert in $cm^2$. To investigate the reproducibility in TEER values from various canine colonoid-derived monolayers, TEER measurement was performed in 2 biological replicates with 4 technical replicates using 3 different canine colonoid lines (S3 Fig). To assess the effect of culture conditions on TEER, the colonoid-derived monolayer was cultured with proliferation medium (complete medium with Wnt3a proteins) or differentiation medium (a complete medium without Wnt3a) after forming a monolayer which was at Day 4. This study was performed in 2 biological replicates with 4 technical replicates in each condition (i.e., Diff vs. Control). The medium in the Transwell insert was changed to either differentiation medium or proliferation medium while the bottom wells were filled with proliferation medium.

To assess intestinal barrier permeability, fluorescein sodium salt ($M_w$, 376.27 Da; 0.05 μg/mL) was used as a paracellular marker. The concentration of fluorescein that transported through the cell monolayer (from apical to basolateral) was measured by SpectraMax microplate reader (Molecular Devices). The apparent permeability ($P_{app}$) was calculated using the following equation: $P_{app} = (dQ/dt)/(C_0 \times A)$, where dQ/dt (μg/sec) is the steady-state flux, $C_0$ (μg/mL) is the initial concentration of the fluorescein in the apical chamber, and A ($cm^2$) is the surface area cultured on the nanoporous insert. This experiment was performed with 2 biological and 4 technical replicates.

## Immunofluorescence imaging

For IF microscopic analysis, a confluent cell monolayer grown on a nanoporous insert was fixed with 4% (w/v) paraformaldehyde (Electron Microscopy Science) for 15 min at room temperature. Samples were then permeabilized with 0.3% (v/v) Triton X-100 (Sigma) and blocked

with 2% (w/v) bovine serum albumin (BSA; Sigma) followed by PBS ($Ca^{2+}$ and $Ma^{2+}$ free; Gibco) washing. The monolayer was incubated at room temperature for 1 h with primary antibodies against ZO-1 (Invitrogen), P-gp (Thermo Fisher Scientific), CgA (Abcam), and Ki67 (Abcam) diluted in 2% (w/v) BSA in PBS. Alexa Fluor 488 conjugated E-cadherin (BD Biosciences) was applied in a same procedure. Secondary antibodies of Alexa Fluor 555-conjugated goat polyclonal anti-rabbit IgG (Abcam) for ZO-1, P-gp, CgA, and Ki67 diluted in 2% (w/v) BSA in PBS were applied under light protected conditions at room temperature for 1 h. For the counterstaining, samples were incubated with 4',6-diamidino-2-phenylindole dihydrochloride (DAPI) (1 μg/mL; Fisher Scientific) and Alexa Fluor 647-conjugated phalloidin (7.5 units; Thermo Fisher) for nuclei and F-actin visualization, respectively. To detect the mucus production on the monolayer, samples were directly stained with Alexa Fluor 488-conjugated WGA (5.0 μg/mL; Thermo Fisher). The monolayer was imaged using a differential interference contrast (DIC) or laser-scanning confocal microscopy (DMi8; Leica). Acquired images were processed using LAS X (Leica) or ImageJ v1.52q [58]. The percentage of cell numbers (Ki67 and CgA) or fluorescence intensity (P-gp, ZO-1, and E-cad) was assessed using ImageJ to the randomly selected images that show representative characteristics. The number of cells that show positive signals was manually counted (ImageJ), then the number was normalized by the total number of nuclei to calculate the % population. For this quantification, 3 independent fields of view from 4 independent biological replicates were used, while at least two technical replicates were performed (Fig 3F). For the quantification of the P-gp expression, total 10 randomly chosen fields of view to detect P-gp expression levels among 4 biological replicates, while at least two technical replicates were performed (Fig 5C). For the quantitative assessment of ZO-1 and E-cadherin, total 10 and 6 randomly chosen fields of view for ZO-1 and E-cadherin, respectively, to quantify the relative intensity of fluorescence among 4 biological replicates of IF staining experiment. We also applied two technical replicates to the individual biological replicate. (S2 Fig).

## *In situ* hybridization of mRNA

We employed RNA-ISH using the RNAscope Multiplex Fluorescent Reagent Kit v2 (Advanced Cell Diagnostic, Newark, CA) [55] on a canine colonoid-derived monolayer to characterize the multi-lineage differentiation. In brief, a colonoid-derived monolayer was fixed and underwent dehydration/hydration, permeabilization, and protease treatment. Samples were hybridized in the ACD HybEZ II Hybridization System (110v) oven at 40˚C while placed in light protected humidified trey as instructed by the manufacture [55]. The samples were then stained for specific oligonucleotide probes for visualizing intestinal stem cells (CL-Lgr5-C2; Advanced Cell Diagnostic), differentiated intestinal epithelial cells (Cl-ALPI; Advanced Cell Diagnostic), and secretory enteroendocrine cells (Cl-NEUROG3-C3; Advanced Cell Diagnostic), respectively. Next, amplification and visualization using Opal 520 (FP1487001KT), Opal 570 (FP1488001KT), and Opal 650 (FP1496001KT) were performed. Sections were imaged using a confocal microscope (DMi8; Leica). Acquired images were processed using LAS X (Leica) or ImageJ. The number of cells staining positive for mRNA detection for each RNAscope probe was manually counted at random positions. Specifically, the number of cells staining positive was manually counted, then normalized by the total number of nuclei. Quantification of the positive cells to individual RNA markers was performed with 3 independent fields of view from 2 independent biological replicates (Fig 3F). Probes against RNA Polymerase II Subunit A (POLR2A) and Ubiquitin C (UBC) were applied and the same amplification and visualization steps were performed to prepare the positive control (S1 Fig).

## Transmission and scanning electron microscopy

After 13 days of culture, the culture medium was gently removed from the apical and basal chambers of the Transwell, and cells were fixed with 2% (v/v) glutaraldehyde (Electron Microscopy Sciences) in 0.1 M cacodylate buffer (Electron Microscopy Sciences) for 1 hr at room temperature, and washed in 0.1 M cacodylate buffer. Samples were then fixed and stained with 1% (v/v) osmium tetroxide (Electron Microscopy Sciences) and 1% (v/v) ferrocyanide in cacodylate buffer, and then stained with 2% (v/v) uranyl acetate for a negative contrast. Samples were finally dehydrated through serial dehydration in ethanol from 50% to 100% (v/v) and then infiltrated with resin (Electron Microscopy Sciences) to be polymerized at 60°C and sectioned for TEM. Ultrathin (50–100 nm) sections were cut by a microtome with a diamond blade, then collected on copper grids and observed under the Transmission Electron Microscope (FEI Tecnai) using an accelerating voltage of 80 kV. SEM samples were fixed in 2.5% (v/v) glutaraldehyde (Electron Microscopy Sciences), treated with 1% (v/v) osmium tetroxide (Electron Microscopy Sciences) in 0.1 M sodium cacodylate buffer (Electron Microscopy Sciences) for 30 min at room temperature. Samples were dehydrated through serial dehydration in ethanol from 50% to 100%, and hexamethyldisilazane (HDMS) method. Samples were coated with a thin (12 nm) layer of Pt/Pd using a sputter coater (Cressington 208 Benchtop Sputter) prior to imaging using an SEM (Zeiss Supra 40V SEM) with an accelerating voltage of 5 kV. The average frequencies of microvilli in less frequent and frequent areas were performed at 4 random independent positions from 3 different SEM images.

## Statistical analysis

All results are expressed as mean±standard error (SEM). Shapiro-Wilk tests were used to assess the normality of the data. Mann–Whitney U test (for non-parametric data) or student's t-tests (for parametric data) were used to compare the expression levels of proteins between two different time points (Day 3 vs. Day 13), TEER and $P_{app}$ values on different culture time points (Day 2 vs. Day 6), or TEER values in different culture conditions (proliferation medium vs. differentiation medium) at each culture time point. All statistical analyses were performed using Prism 8.2.1 (GraphPad Software, San Diego, CA). $P$ values $< 0.05$ were considered statistically significant.

## Supporting information

**S1 Fig. Verification of the RNA *in situ* hybridization on the canine colonoid-derived epithelial monolayer.** A 3-Plex Positive Control Probe (Advanced Cell Diagnostics) was applied to the canine monolayer cultured for 13 days to confirm the functionality of the kit applied. A low (RNA Polymerase II Subunit A (POLR2A), Opal 650; **S1A**) and a high expressor RNA (Ubiquitin C (UBC), Opal 520; **S1B**) confirmed the functionality of the probes applied in the canine epithelial monolayer. An overlaid image is displayed in **S1C**. Nuclei, blue. Bars, 50 μm. (TIF)

**S2 Fig. Expression of the epithelial junctional proteins in the canine colonoid-derived epithelial monolayer.** Quantification of the expression level of ZO-1 and E-cadherin at days 3 and 13 was performed using total 10 and 6 randomly chosen fields of view for ZO-1 and E-cadherin, respectively, among 4 biological replicates of IF staining experiment. We also applied two technical replicates to individual biological replicates. a.u., arbitrary unit. NS, not significant. (TIF)

**S3 Fig. Reproducibility of the barrier function of colonoid-derived epithelial monolayers derived from three different canine colonoid lines.** Three independent lines of canine colonoids show similar profile of epithelial barrier function when those three lines were used to form a monolayer on a nanoporous insert. The result was produced with 2 biological replicates, where each biological replicate was performed with 4 technical replicates. Error bars indicate SEM.
(TIF)

## Acknowledgments

We thank Mrs. Michelle A. Mikesh (Center for Biomedical Research Support Microscopy & Imaging Core Facility, UT Austin) and Dr. Sonali Jog (Advanced Cell Diagnostics) for their technical support.

## Author Contributions

**Conceptualization:** Yoko M. Ambrosini, Albert E. Jergens, Karin Allenspach, Jonathan P. Mochel, Hyun Jung Kim.

**Data curation:** Yoko M. Ambrosini, Yejin Park, Woojung Shin, Soyoun Min.

**Formal analysis:** Yoko M. Ambrosini, Yejin Park.

**Funding acquisition:** Yoko M. Ambrosini, Yejin Park, Albert E. Jergens, Woojung Shin, Jinah Jang, Karin Allenspach, Jonathan P. Mochel, Hyun Jung Kim.

**Investigation:** Yoko M. Ambrosini, Yejin Park, Woojung Shin, Soyoun Min.

**Methodology:** Yoko M. Ambrosini.

**Project administration:** Yejin Park, Hyun Jung Kim.

**Resources:** Yoko M. Ambrosini, Yejin Park, Albert E. Jergens, Woojung Shin, Todd Atherly, Dana C. Borcherding, Karin Allenspach, Jonathan P. Mochel, Hyun Jung Kim.

**Software:** Hyun Jung Kim.

**Supervision:** Hyun Jung Kim.

**Validation:** Yoko M. Ambrosini.

**Visualization:** Yoko M. Ambrosini, Yejin Park, Hyun Jung Kim.

**Writing – original draft:** Yoko M. Ambrosini, Yejin Park.

**Writing – review & editing:** Yoko M. Ambrosini, Yejin Park, Albert E. Jergens, Karin Allenspach, Jonathan P. Mochel, Hyun Jung Kim.

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
