## [Decision Letter · Decision Letter 0]

23 Jan 2020

PONE-D-20-00243

Recreation of an Accessible Interface of the Biopsy-Derived Canine Intestinal Organoids to Study Epithelial-Luminal Interactions

PLOS ONE

Dear Kim,

Thank you for submitting your manuscript to PLOS ONE. After careful consideration, we feel that it has merit but does not fully meet PLOS ONE’s publication criteria as it currently stands. Therefore, we invite you to submit a revised version of the manuscript that addresses all the points raised during the review process.

Two experts have evaluated the manuscript. Both of them agreed that the work is valuable. The novel aspects should be better described with appropriate citations of the literature for models from other species. Overstatements should be avoided and additional  validation of the model is needed together with amendments related to methodological issues. 

We would appreciate receiving your revised manuscript by Mar 08 2020 11:59PM. To enhance the reproducibility of your results, we recommend that if applicable you deposit your laboratory protocols in protocols.io, where a protocol can be assigned its own identifier (DOI) such that it can be cited independently in the future. For instructions see: http://journals.plos.org/plosone/s/submission-guidelines#loc-laboratory-protocols

We look forward to receiving your revised manuscript.

Kind regards,

Mária A. Deli, M.D., Ph.D.

Academic Editor

PLOS ONE

Journal Requirements:

'J.P.M., A.J., K.J.A., and H.J.K. are co-founders of 3D Health Solutions Inc. and hold an equity interest in the company. T.A. is employed by 3D Health Solutions Inc.'

Reviewers' comments:

Reviewer's Responses to Questions

**Comments to the Author**

1. Is the manuscript technically sound, and do the data support the conclusions?

Reviewer #1: Partly

Reviewer #2: Partly

2. Has the statistical analysis been performed appropriately and rigorously? 

Reviewer #1: Yes

Reviewer #2: I Don't Know

3. Have the authors made all data underlying the findings in their manuscript fully available?

Reviewer #1: Yes

Reviewer #2: Yes

4. Is the manuscript presented in an intelligible fashion and written in standard English?

Reviewer #1: Yes

Reviewer #2: Yes

5. Review Comments to the Author

Reviewer #1: Ambrosini et al present a method to generate epithelial monolayers from canine intestinal organoids. Although this is the first study of this kind generating canine intestinal monolayers, the methods used to generate these monolayers are not novel, asides from slight variations in the dissociation and plating protocols/media, having been demonstrated by multiple groups in both mouse and human systems. They suggest that these monolayers form epithelial cell layers with multiple differentiated cell types, including enteroendocrine cells, intestinal stem cells and polarised enterocytes. Although some of the data supports this conclusion, such as the TEM images of polarised cells and the results from the TEER experiments, there is a lack of data to show fully that these cells are properly differentiated into functioning intestinal epithelium. Some of the markers used are not appropriate and it is not clear from the select number of TEM images presented whether goblet cells are present and whether there are true microvilli on the apical surface of these cells as has been observed in human and murine organoid culture. Furthermore, it is not clear in the manuscript how many canine lines were used in these experiments. The authors suggest in their discussion that this protocol “could potentially be used for translational Precision Medicine purposes because the individual canine organoid lines obtained from various dog species can be established and utilized for pharmaceutical studies”. However, it is not clear that multiple organoid lines were used in this study, limiting the potential applicability for the future uses of the system the authors suggest. Overall, although I agree with the authors that canine organoids would provide a useful system for various areas of research, I don’t believe that the data presented here is enough to demonstrate successful monolayer development.

Major comments

1. In my opinion, the authors do not convincingly show that there are multiple fully differentiated cell types present in their canine epithelial model system. % positive cells presented in figure 2F is not showing distinct cell types because if it was the total % would equal 100, whereas the number shown is much greater than this. One reason may be the assumption that differentiated enteroendocrine cells can be identified by expression of Neurog3, whereas Neurog3 is expressed in progenitor cells, which become mature enteroendocrine cells but also other cell types (see Gehart et al., 2019. Cell: 176(5)). To identify mature enteroendocrine cells it is more common to use markers such as ChgA and Reg4 to define mature cells of this subset. Although the limitations of using Neurog3 are discussed by the authors in their discussion, it is not clear why the authors not used one of these more common enteroendocrine markers? The authors have used transmission electron microscopy to nicely show the presence of tight junctions between epithelial cells in Figure 3, therefore if there are fully differentiated epithelial cell subsets such as goblet cells and enteroendocrine cells these should be identifiable based on distinct morphology in comparison to absorptive enterocytes by TEM (see Figure S1, Forbester et al., 2018. PNAS: 115(40)). As the authors point out, the availability of canine-specific antibodies is a limitation of this study, however TEM would allow them to convincingly show varying morphology between different cell types, without requiring antibodies. The reliance on the results from the FISH experiments, which are difficult to interpret, is not convincing enough data. Therefore, the statement in the discussion: “Canine epithelium cultured on a Transwell insert differentiated into multiple lineages of the differentiated intestinal epithelium (e.g., intestinal stem cells, absorptive enterocytes, and enteroendocrine cells)” is not validated by enough convincing data within the manuscript.

2. The authors imply that one advance of their study is that they use enzymatic rather than mechanical disruption to disrupt the organoid ultrastructure to single cells. This is not a novel method for generating monolayers, and has been used to dissociate and grow monolayers in both and mouse and human intestinal organoid systems (see Altay et al., 2019. Scientific reports; 9(10140) and Thorne et al., 2018. Developmental Cell; 44(5)). This needs to be clarified in text, because at the moment in my opinion the novelty is overstated. However, in mouse and human systems extensive mechanical disruption is needed in conjunction to TrypLE treatment to ensure dissociation to single cells. Can the authors explain why dissociation of canine intestinal organoids is much easier in comparison to mouse and human intestinal organoids?

3. The microvilli in Figure 3A and Figure 4E look disrupted, or more similar to the structures seen on the surface of M-cells? Can the authors explain why this is? Why is only a single ‘normal length’ villus shown in Figure 3C? See Llanos-Chea et al., 2019. J Pediatr Gastroenterol Nutr; 68(4) for sample microvilli on human intestinal epithelial organoids.

Minor comments

1. None of the figures specify number of replicates/number of canine intestinal organoid lines used. Are these figures representative for lines from multiple canine donors?

2. Description in figure legend for Figure 4D is confusing, needs to be clearer what the control and the experimental samples are, the sentence doesn’t make sense

3. Figure 2F – not clear what N=3 is, 3 fields of view of the same sample; 3 replicate monolayers and staining experiments from the same canine organoid line; 3 replicate monolayers and staining experiments from different canine organoid lines?

4. Figure 3F- As above what is N equivalent to? Multiple experiments, multiple replicates, multiple organoid lines?

5. Figure 4C, D & G – as above, what does N represent in terms of replicates?

Reviewer #2: In this manuscript, Ambrosini and colleagues describe the culture of canine cells derived from 3D canine colonic organoids as monolayers. Various measures of cellular differentiation and monolayer integrity are described. The study is meant as follow up of previously published work (ref 9 in this manuscript) and in part overlapping statements in introduction and discussion are used to justify the need for development of canine culture models. Also TrypLE express is commonly used for dissociation of colonic 3d and 2d intestinal cultures and cited in multiple papers (Thorne CA, Chen IW, Sanman LE, Cobb MH, Wu LF, Altschuler SJ. Developmental cell. 2018;44(5):624-33.e4, VanDussen KL, Marinshaw JM, Shaikh N, Miyoshi H, Moon C, Tarr PI, et al. Gut. 2015;64(6):911-20…), therefore attributing it as a novel enzymatic approach (line 9,18, 67 and so on…) is simply overstating. Except for “optimization” of cell numbers needed to seed one type/brand of transwell, this study does not significantly contribute to the “reproducible method” either as it is not clear if the 2D lines used are derived from different animals(line 67).

Other Concerns:

Title: Please consider changing “Recreation of accessible interface….” to “ Establishment of accessible interface or Recapitulation of accessible interface….”

Line 60-63 Consider changing the sentence: “Thus, cultures of polarized…..” into: Thus cultures of polarized intestinal cell monolayers are better suited for standardized measure of transepithelial permeability and epithelial-luminal interaction due to easier accessibility of the apical surface.

Methodologic issues:

1. Throughout the paper N is mentioned for each experiment, but not described properly. Does N means 3 biopsies from the same dog or 1 biopsy of three different dogs. In the opinion of the referee in order to contribute to the statement of “standardized protocol” the authors should use different dogs as N. Currently this is not clear Throughout the manuscript. For example: figure 2F, are 3 biopsies from the same animal of are 3 different animals used? Figure 3 D and E , what is N precisely, 10 different donors or 10 different biopsies from the same donor?

2. figure 2: authors performed WGA staining for mucus layer. Which cells are producing the mucus, are there Goblet cells present in canine colonic monolayers?

3. figure 2: it would be useful to show dapi staining. Right now it is difficult to evaluate these staining, for example ALPI seems to be expressed in the nucleus. Besides, magnification insets are too small to provide more detail.

4. In support to figure 4A-D authors should measure permeation by performing FD4 permeation rate and really show the tight barrier already at day 3. This is beneficial to those wishing to use this model in order to decide the right time for an intervention study. The idea for authors is to elucidate it and provide a simple timeline of potential intervention study.

6. PLOS authors have the option to publish the peer review history of their article (what does this mean?). If published, this will include your full peer review and any attached files.

Reviewer #1: No

Reviewer #2: No

---

## [Author Response · Author response to Decision Letter 0]

9 Mar 2020

Responses to Reviewers (PONE-D-20-00243) 

Reviewer 1: 

1. Ambrosini et al present a method to generate epithelial monolayers from canine intestinal organoids. Although this is the first study of this kind generating canine intestinal monolayers, the methods used to generate these monolayers are not novel, asides from slight variations in the dissociation and plating protocols/media, having been demonstrated by multiple groups in both mouse and human systems. They suggest that these monolayers form epithelial cell layers with multiple differentiated cell types, including enteroendocrine cells, intestinal stem cells and polarised enterocytes. Although some of the data supports this conclusion, such as the TEM images of polarised cells and the results from the TEER experiments, there is a lack of data to show fully that these cells are properly differentiated into functioning intestinal epithelium. Some of the markers used are not appropriate and it is not clear from the select number of TEM images presented whether goblet cells are present and whether there are true microvilli on the apical surface of these cells as has been observed in human and murine organoid culture. Furthermore, it is not clear in the manuscript how many canine lines were used in these experiments. The authors suggest in their discussion that this protocol “could potentially be used for translational Precision Medicine purposes because the individual canine organoid lines obtained from various dog species can be established and utilized for pharmaceutical studies”. However, it is not clear that multiple organoid lines were used in this study, limiting the potential applicability for the future uses of the system the authors suggest. Overall, although I agree with the authors that canine organoids would provide a useful system for various areas of research, I don’t believe that the data presented here is enough to demonstrate successful monolayer development:

We appreciate the Reviewer’s positive comment on the promising perspectives of our canine colonoid- derived monolayer system for various areas of research applications. In this research article, we have claimed the value of outcomes in terms of the optimization of the protocol of dissociation, seeding, and culture of canine colonoid-derived epithelium on the standpoint of an enduser for assessing the morphological characteristics, polarization of apical brush border, basic barrier function and permeability, and expression of lineage-dependent or functional markers.

We agree with the Reviewer’s critique in terms of the insufficient novelty of the dissociation method that we adapted in this study to recreate a tissue interface, which has been applied and optimized in mouse, pig, or human intestinal organoids by multiple groups (1,2). Thus, we toned down the overstatement of the “novel” or “standardized” enzymatic dissociation approach to “an optimized protocol” in the revised manuscript in lines of 10, 65, 69, and 157. Instead, we sufficiently claimed the specific “usefulness” and “functionality” of our canine colonoid-derived epithelial monolayer reconstituted on the nanoporous membrane for the experimental assessment including barrier integrity and permeability, localization of the key structural markers, spatial visualization of the stem cells and other colonic cells with lineage-dependent cytodifferentions, and the stability and reproducibility of the formation and maintenance of a primary colonic epithelial monolayer with required statistics.

To improve our argument based on the Reviewer’s comments, we additionally performed morphological characterization using transmission electron microscopy (TEM) and scanning electron microscopy (SEM) to demonstrate the presence of goblet cells in our canine colonoid-derived monolayer. We summarized the new results that specifically support the expression of goblet cells in Figure 4 in the revised manuscript. Here, we identified the mucin granule-containing goblet cells using TEM (revised Fig 4B, “MG”), goblet cell orifices (revised Fig 4D, “GO”), and fenestrated membranes (revised Fig 4D, “FM”) extending deep into the goblet cell surface using SEM. This new finding shows a good agreement with the previous studies (3,4), demonstrating that the goblet cells were present in the canine colonoid-derived monolayer.

Based on the Reviewer’s comments, we re-evaluated the stem and differentiated cells populated on the caine colonoid-derived epithelial monolayer in the revised manuscript using various imaging modalities including additional antibody-based immunofluorescence (IF) staining as well as RNA in situ hybridization. This precision imaging technology with high specificity reveals that our marker staining results show the presence of stem- and differentiated cells that are pertinent in the colonic functions and cytodifferentiation with strong in vivo relevance. In terms of the quantification of epithelial populations, the % positive cell numbers for the Lgr5+ stem cells (5,6), ALPI+ differentiated intestinal epithelium (4,6), Neurog3+ and CgA+ enteroendocrine cells (7,8) were detected as 7.6±0.1, 60.1±0.9, 41.2±10.3%, and 47.8±2.7%, respectively, which are all strongly supported by the previous reports published in both human and dog in vitro studies. Our statistical analysis also supports the reproducibility and robustness of our experimental outcomes in the revised manuscript (see Statistical analysis). 

We used three independent lines of canine colonoids in this study. We additionally inserted a new Supplementary Figure 3 in the revised manuscript to show the reproducibility of the functional outcome in terms of the barrier function using the TEER measurement of the used 3 indepenent colonoid lines with 2 biological replicates and 4 technical replicates. We provided this updated information in both Materials and Methods (lines 131-136) and Discussion (lines 343-346) sections. As a proof-of-principle study, we applied reasonable sets of biological and technical replications (see details in the revised manuscript, especially in the Legend) to claim the potential contribution to the translational researches between canine and human studies. Our manuscript clearly demonstrates the optimized protocol to recreate the mucosal tissue interface using primary canine colonic epithelium for the “future applicability”. The epithelial barrier functions and colon-specific characteristics support that our technical advance may contribute to validate our system toward inter-species variation in barrier permeability, transport, absorption, efflux, or metabolism of the administered drug compounds. We currently perofrm these aspects as follow-up studies; however, we do not include them in this manuscript as they are beyond the scope.

2. In my opinion, the authors do not convincingly show that there are multiple fully differentiated cell types present in their canine epithelial model system. % positive cells presented in figure 2F is not showing distinct cell types because if it was the total % would equal 100, whereas the number shown is much greater than this. One reason may be the assumption that differentiated enteroendocrine cells can be identified by expression of Neurog3, whereas Neurog3 is expressed in progenitor cells, which become mature enteroendocrine cells but also other cell types (see Gehart et al., 2019. Cell: 176(5)). 

In terms of the interpretation of our data, the Reviewer’s argument of “if it was the total % would equal 100, whereas the number shown is much greater than this” is incorrect because Ki67+ cells are not the population of lineage-dependent cells. Thus, we do not expect that the add-up of the provided chart would be 100%. In the revised manuscript and figures, we provide additional marker staining results (Chromogranin A, CgA) as well as the quantification (revised Fig 3). In terms of the quantification of epithelial populations, the % positive cell numbers for the Lgr5+ stem cells (5,6), ALPI+ differentiated intestinal epithelium (4,6), Neurog3+ and CgA+ enteroendocrine cells (7,8) were detected as 7.6±0.1, 60.1±0.9, 41.2±10.3%, and 47.8±2.7%, respectively, which are all strongly supported by the previous reports published in both human and dog in vitro studies. Our statistical analysis also supports the reproducibility and robustness of our experimental outcomes in the revised manuscript (see Statistical analysis). 

In order to demonstrate the mature enteroendocrine cell differentiation and its % positive cell number, we performed an IF staining against canine Chromogranin A, which actually showed similar % expression level as Neurog3 as well as a reference from a human large intestinal tissue (7). For this quantification, three independent fields of view from two or more independent biological replicates were used. We also applied at least two technical replicates (revised Fig 3F). We appreciate the suggestion of references by the Reviewer.

3. To identify mature enteroendocrine cells it is more common to use markers such as ChgA and Reg4 to define mature cells of this subset. Although the limitations of using Neurog3 are discussed by the authors in their discussion, it is not clear why the authors not used one of these more common enteroendocrine markers? 

As suggested, we performed IF staining against canine Chromogranin A (CgA) in the revised manuscript and shown in Fig 3E. We updated the Results (lines 105-111), and Materials and Methods (lines 366-381) sections accordingly. 

4. The authors have used transmission electron microscopy to nicely show the presence of tight junctions between epithelial cells in Figure 3, therefore if there are fully differentiated epithelial cell subsets such as goblet cells and enteroendocrine cells these should be identifiable based on distinct morphology in comparison to absorptive enterocytes by TEM (see Figure S1, Forbester et al., 2018. PNAS: 115(40)). As the authors point out, the availability of canine-specific antibodies is a limitation of this study, however TEM would allow them to convincingly show varying morphology between different cell types, without requiring antibodies.

In response to the comment for the verification of goblet cells, we performed SEM and additional TEM image acquisitions to demonstrate the presence of goblet cells in our canine organoid-derived monolayer. Our new findings are now summarized in Fig 4 in the revised manuscript. Briefly, we identified the mucin granule-containing goblet cells using TEM (revised Fig 4B, “MG”). In addition, we leveraged the SEM iamge results to identify the goblet cells, where goblet cell orifices (revised Fig 4D, “GO”) and fenestrated membranes (revised Fig 4D, “FM”) were clearly indicated. This observation shows a strong agreement with the previous studies (3,4), demonstrating that the goblet cells were present in the canine colonoid-derived monolayer. In response to the comment for the verification of enteroendocrine cells, we performed IF staining against canine Chromogranin A in addition to the visualization using RNA in situ hybridization of Neurog3+ cells and summarized in Fig 3E and 3F in the revised manuscript. We updated the Results (lines 105-111) and Materials and Methods (lines 366-381) sections accordingly. We appreciate the suggestion of references by the Reviewer.

5. The reliance on the results from the FISH experiments, which are difficult to interpret, is not convincing enough data. 

We understand the Reviewer’s concern in terms of the obviousness and easiness of the data interpretation of the RNA in situ hybridization results. However, we believe that our RNA in situ hybridization data successfully provide a localized marker-positive cell that highlights the expression of mRNA. Our argument is supported by the following reasons. First, RNA in situ hybridization specifically show the presence and its location of the target RNA, not protein, by which the marker-positive cell show the “on” signal via fluorophore. Thus, the signal may be somewhat weak with a form of “dots” (revised Figs 3A, 3C, 3D, and S1) rather than the positioned “space” in fluorescence (revised Fig 3B and 3E). The displayed signal of collected “dots” may position on the nucleus or the cytoplasm, but it does not compromise the evidence that the cell shows the targeted fluorescence signal. Second, although RNA in situ hybridization is a relatively new technology, the validation and reproducibility of this technology have been sufficiently established and proved by multiple studies with in vitro intestinal models (9–13). Third, we previously provided the successful demonstration of the RNA in situ hybridization on the canine intestinal 3D organoid cultures, where we specifically visualized stem cells (Lgr5, Sox9, and Ephb2), paneth-like cells (Fzd5 and Cath), absorptive enterocytes (ALPI), and enteroendocrine cells (Neurog3) (6). Finally, we provided positive control results in Supplementary Fig 1 to confirm the appropriate performance of the RNA in situ hybridization in the canine colonoid-derived monolayer, verifying that the positive and negative staining provide the precision and accuracy of the RNA in situ hybridization that targets the genes that are express in a lower (revised Fig S1A) and higher degree (revised Fig S1B) in the cells.

6. The authors imply that one advance of their study is that they use enzymatic rather than mechanical disruption to disrupt the organoid ultrastructure to single cells. This This is not a novel method for generating monolayers, and has been used to dissociate and grow monolayers in both and mouse and human intestinal organoid systems (see Altay et al., 2019. Scientific reports; 9(10140) and Thorne et al., 2018. Developmental Cell; 44(5)). This needs to be clarified in text, because at the moment in my opinion the novelty is overstated. However, in mouse and human systems extensive mechanical disruption is needed in conjunction to TrypLE treatment to ensure dissociation to single cells. Can the authors explain why dissociation of canine intestinal organoids is much easier in comparison to mouse and human intestinal organoids?

We agree with the Reviewer’s critique in terms of the insufficient novelty of the dissociation method that we adapted in this study to recreate a tissue interface, which has been applied and optimized in mouse or human intestinal organoids by multiple groups (1,2). Thus, we toned down the overstatement of the “novel” or “standardized” enzymatic dissociation approach to “an optimized protocol” in the revised manuscript in lines of 10, 65, 69, and 157. Instead, we sufficiently claimed the specific “usefulness” and “functionality” of our canine colonoid-derived epithelial monolayer reconstituted on the nanoporous membrane for the experimental assessment including barrier integrity and permeability, localization of the key structural markers, spatial visualization of the stem cells and other colonic cells with lineage-dependent cytodifferentions, and the stability and reproducibility of the formation and maintenance of a primary colonic epithelial monolayer with required statistics. It is noted that we did not claim “dissociation of canine intestinal organoids is much easier in comparison to mouse and human intestinal organoids”. For the Reviewer’s information, we have applied the same dissociation protocol to both human and canine intestinal organoids, where we have not observed a significant difference in the dissociation yield. We discussed this point in the revised manuscript (lines 158-160). We appreciate the suggestion of references by the Reviewer. 

7. The microvilli in Figure 3A and Figure 4E look disrupted, or more similar to the structures seen on the surface of M-cells? Can the authors explain why this is? Why is only a single ‘normal length’ villus shown in Figure 3C? See Llanos-Chea et al., 2019. J Pediatr Gastroenterol Nutr; 68(4) for sample microvilli on human intestinal epithelial organoids?

In the revised manuscript, we provide new data sets performed with the SEM imaging. Briefly, SEM images demonstrate the overall patterns of the recreated microvilli on the apical surface of our canine colonoid-derived monolayer, confirming the frequency and density of the microvilli (revised Fig 2). The variation in microvilli frequency was observed in our dog colonoid-derived monolayer (revised Fig 2B), which was also noted in other colonoid-derived studies (14,15). The number of microvilli assessed by the SEM imaging was variable in the range from 9 to 18 microvilli/µm2, which was similar to the report of human intestinal epithelial organoid culture performed in vitro organ-on-a-chip (16).

In terms of a “single normal length (micro)villus”, we removed that image from the figure because we confirm that it is not representative based on the investigation of SEM images. Stunted microvilli were observed in our system which could reflect the fact that colonic intestinal cells may not require longer microvilli due to minimal nutrient absorption in the colon (17). Possibly, it could be due to the culture condition that is not completely adequate to promote longer microvilli (4). Although assessing the effect of different culture conditions on dog colonoid-derived monolayer, particularly to the length of microvilli, is beyond our scope in this manuscript, such investigation would be beneficial to better understand the physiological demonstration and functions of the microvilli in the future study. We addressed this point in the revised manuscript (lines 272-282). We appreciate the suggestion of references by the Reviewer. 

Minor comments

8. None of the figures specify number of replicates/number of canine intestinal organoid lines used. Are these figures representative for lines from multiple canine donors?

The number of replicates and the number of canine intestinal organoid lines were specifically delineated in each figure lengend as well as in the Materials and Methods section (lines 383-391, lines 408-412) in the revised manuscript. As previously mentioned, we used three independent lines of canine colonoids derived from three independent canine donor biopsies. Based on the provided biological and technical replicates, we confirm that the figures are representative for lines from multiple canine donors. 

9. Description in figure legend for Figure 4D is confusing, needs to be clearer what the control and the experimental samples are, the sentence doesn’t make sense

We provided a new legend for Fig 4D (revised Fig 6C) in the revised manuscript as suggested with improved clarity, accuracy, and brevity. 

10. Figure 2F – not clear what N=3 is, 3 fields of view of the same sample; 3 replicate monolayers and staining experiments from the same canine organoid line; 3 replicate monolayers and staining experiments from different canine organoid lines?

The N number indicates the number of a field of view. In Fig 2F (revised Fig 3F), we chose three independent fields of view from two or more independent biological replicates. We also applied at least two technical replicates to all the experimental setup. The image was randomly chosen for the analysis. The described information is now specifically provided in the figure legend in the revised manuscript.

11. Figure 3F- As above what is N equivalent to? Multiple experiments, multiple replicates, multiple organoid lines?

In Figure 3F (revised Figure 5C), we used total 10 randomly chosen fields of view to detect P-gp expression levels among 4 biological replicates. In each biological replicate, we performed 2 technical replicates. The described information is now specifically provided in the figure legend in the revised manuscript.

12. Figure 4C, D & G – as above, what does N represent in terms of replicates?

In both Fig 4C and 4D (revised Supplementaary Fig 2), we used total 10 and 6 randomly chosen fields of view for Fig 4C and 4D, respectively, to quantify the relative inteinsity of fluorescence among 4 biological replicates of IF staining experiment. We also applied two technical replicates to individual biological replicates. In Fig 4G (revised Fig 6C), TEER values were caculated from total eight replicates (N=8), where 2 biological replicates with 4 technical replicates in each condition were applied for the TEER assessment in each data point. 

Reviewer 2: 

1. The study is meant as follow up of previously published work (ref 9 in this manuscript) and in part overlapping statements in introduction and discussion are used to justify the need for development of canine culture models. Also, TrypLE express is commonly used for dissociation of colonic 3d and 2d intestinal cultures and cited in multiple papers (Thorne CA, Chen IW, Sanman LE, Cobb MH, Wu LF, Altschuler SJ. Developmental cell. 2018;44(5):624-33.e4, VanDussen KL, Marinshaw JM, Shaikh N, Miyoshi H, Moon C, Tarr PI, et al. Gut. 2015;64(6):911-20…), therefore attributing it as a novel enzymatic approach (line 9,18, 67 and so on…) is simply overstating. Except for “optimization” of cell numbers needed to seed one type/brand of transwell, this study does not significantly contribute to the “reproducible method” either as it is not clear if the 2D lines used are derived from different animals (line 67).

We agree with the Reviewer’s critique in terms of the insufficient novelty of the dissicoation method that we adapted in this study to recreate a tissue interface, which has been applied and optimized in mouse or human intestinal organoids by multiple groups (1,2). Thus, we toned down the overstatement of the “novel” or “standardized” enzymatic dissociation approach to “an optimized protocol” in the revised manuscript in lines of 10, 65, 69, and 157. Instead, we sufficiently claimed the specific “usefulness” and “functionality” of our canine colonoid-derived epithelial monolayer reconstituted on the nanoporous membrane for the experimental assessment including barrier integrity and permeability, localization of the key structural markers, spatial visualization of the stem cells and other colonic cells with lineage-dependent cytodifferentions, and the stability and reproducibility of the formation and maintenance of a primary colonic epithelial monolayer with required statistics. In terms of the argument of “reproducible method”, we provided repeated, consistent methods of statistics to all our provided data set. In terms of the reproducibility of the efficiency of monolayer generation, it can be influenced by the quality of colonoid (e.g., original viability), yield of dissociation into single cells, applied protocols (mechanical vs. enzymatic), duration of time performed, and incubation condition, except for “optimization” of cell numbers needed to seed one type/brand of Transwell. We used three independent lines of canine colonoids derived from the biopsies of three different canine donors. In terms of the data analysis, we used at least two independent biological replicates with at least two technical replicates, where the imaging data were acquired from at least three different locations. The person-to-person variation of the yield of culture performance as well as the variability of results between batches were also carefully considered to implement the “reproducible method” in this study. Now, we all specifically restate in the revised manuscript in the main text as well as in the Methods section. 

2. Title: Please consider changing “Recreation of accessible interface….” to “Establishment of accessible interface or Recapitulation of accessible interface….”

Line 60-63 Consider changing the sentence: “Thus, cultures of polarized…..” into: Thus cultures of polarized intestinal cell monolayers are better suited for standardized measure of transepithelial permeability and epithelial-luminal interaction due to easier accessibility of the apical surface.

Based on the Reviewer’s comment, the title was changed to “Recapitulation of the Accessible Interface of Biopsy-Derived Canine Intestinal Organoids to Study Epithelial-Luminal Interactions” in the revised manuscript. The sentence that the Reviewer pinpointed was changed as suggested in the revised manuscript as follows; “Thus, cultures of a polarized intestinal cell monolayer are better suited for the standardized measurement of transepithelial permeability and epithelial-luminal interaction due to easier accessibility of the apical surface.” (revised Lines 61-64).

Methodologic issues:

3. Throughout the paper N is mentioned for each experiment, but not described properly. Does N means 3 biopsies from the same dog or 1 biopsy of three different dogs. In the opinion of the referee in order to contribute to the statement of “standardized protocol” the authors should use different dogs as N. Currently this is not clear Throughout the manuscript. For example: figure 2F, are 3 biopsies from the same animal of are 3 different animals used? Figure 3 D and E , what is N precisely, 10 different donors or 10 different biopsies from the same donor?

The N number indicates the number of a field of view. In Fig 2F (revised Fig 3F), we chose three independent fields of view from two or more independent biological replicates. We also applied at least two technical replicates to all the experimental setup. The image was randomly chosen for the analysis. The described information is now specifically provided in the figure legend in the revised manuscript. As previously mentioned, we used three independent lines of canine colonoids that were derived from different canine donors. 

For Figure 3D and 3D (revised Figure 5A and 5B) were selected from 4 biological replicates of experiments. We also applied at least two technical replicates to all the experimental setup. In Figure 3F (revised Figure 5C), we used total 10 randomly chosen fields of view to detect P-gp expression levels among 4 biological replicates. In each biological replicate, we performed 2 technical replicates. The described information is now specifically provided in the figure legend in the revised manuscript.

As stated, we used three independent lines of canine colonoids, where scientifically rigorous multiple biological and technical replicates were applied in this study to increase the statistical significance. However, we agree with the Reviewer’s point in terms of the limited size of the canine cohort, thus, we toned down the overstatement of the “standardized protocol” to “an optimized protocol” in the revised manuscript. 

4. figure 2: authors performed WGA staining for mucus layer. Which cells are producing the mucus, are there Goblet cells present in canine colonic monolayers?

For the verification of goblet cells, we performed SEM and additional TEM image acquisitions to demonstrate the presence of goblet cells in our canine organoid-derived monolayer. All this points are now summarized in Fig 4 in the revised manuscript. Briefly, we identified the mucin granule-containing goblet cells using TEM (revised Fig 4B, “MG”). In addition, we leveraged the SEM iamge results to identify the goblet cells, where goblet cell orifices (revised Fig 4D, “GO”) and fenestrated membranes (revised Fig 4D, “FM”) were clearly indicated in the revised manuscript. This observation shows a strong agreement with the previous studies (3,4), demonstrating that the goblet cells were present in the canine colonoid-derived monolayer. In addition to the verification of goblet cells using electron microscopy, we also performed immunofluorescence confocal microscopy to confirm the actual mucus production on the colonoid-derived epithelial monolayer. As the WGA staining showed, the mucus-like molecules such as N-acetyl-D-glucosamine is present on the apical side of the canine colonoid-derived monolayer (revised Fig 4). Unfortunately, there are no commercially available antibodies against goblet cell marker (e.g., canine MUC2). Thus, we used the WGA live-cell staining to visually characterize the mucus production as we previously used this marker at various studies (18,19). We revised the Results (lines 112-116), Discussion (lines 196-197), Materials and Methods (lines 375-3768), and Figure 4 Legend sections accordingly.

5. figure 2: it would be useful to show dapi staining. Right now it is difficult to evaluate these staining, for example ALPI seems to be expressed in the nucleus. Besides, magnification insets are too small to provide more detail.

The ALPI staining provided in previously Figure 2C (revised Figure 3C) had the visualized nuclei stained with DAPI and pseudo-colored to grey. We provided this information in figure legend. In terms of the localization of the fluorescence signal of ALPI, RNA in situ hybridization specifically stains RNAs, which commonly exist in both nucleus and cytoplasm. Thus, the positive signals observed in the nucleus may be the overlay of the RNA signal dots above the nucleus or truly the positive signal within the nucleus. Regardless of the location of the positive signal, that is what the probe is detecting and should be interpreted as the way that the positive signal (either a single dot or a collection of dots) is indicative of the presence of the target gene(s) in that particular cell. This RNA in situ hybridization technology has been successfully applied in dog organoids by our group (6) and similar findings (i.e., positive signals seems to be expressed in the nucleus) can be found in other studies (11,13) as well as our positive control provided in Supplementary Fig 1. We increased the magnification insets to avoid any difficulty in data interpretation. We reflected all the discussion here in the revised manuscript. 

6. In support to figure 4A-D authors should measure permeation by performing FD4 permeation rate and really show the tight barrier already at day 3. This is beneficial to those wishing to use this model in order to decide the right time for an intervention study. The idea for authors is to elucidate it and provide a simple timeline of potential intervention study.

In response to this comment, we performed a transport assay using fluorescein sodium salt (MW, 376.27 Da) as a small molecule paracellular fluorescent marker for estimating the apparent permeability (Papp) value. The new results are shown in Figure 6F in the revised manuscript and each data point was prepared with 2 biological and 4 technical replicates. We used fluorescein instead of FD4 because this molecule will be more specific to quantitatively compared the Papp profile as the TEER value showed pretty high level (>1,000 Ohm cm2) after day 4 (not day 3), suggesting that any medium- or large-sized fluorescent paracellular markers may not penetrate through the formed monolayer. Based on the TEER profile, we anticipated that the permeability barrier integrity will be matured and stabilized after day 4 (revised Fig 6C and 6F), whereas morphologically the monolayer showed reasonable tight junctions before day 4 (revised Figs 1C and 5A). Thus, we further confirmed if the day 4 may be a good temporal indicator to recognize the maturity of a colonoid-derived monolayer applied in our study for the functional assessment. We confirmed by applying the fluorescein transport assay, where the inverse correlation was observed compared to the TEER profile (revised Fig 6F) with statistical significance (P <0.0001). We revised our manuscript based on the discussion provided here. 

References: 

1. van der Hee B, Loonen LMP, Taverne N, Taverne-Thiele JJ, Smidt H, Wells JM. Optimized procedures for generating an enhanced, near physiological 2D culture system from porcine intestinal organoids. Stem Cell Res. 2018 Apr 1;28:165–71. 

2. Shin W, Hinojosa CD, Ingber DE, Kim HJ. Human Intestinal Morphogenesis Controlled by Transepithelial Morphogen Gradient and Flow-Dependent Physical Cues in a Microengineered Gut-on-a-Chip. iScience. 2019 May;15:391–406. 

3. Marsh MN, Swift JA. A study of the small intestinal mucosa using the scanning electron microscope. Gut. 1969 Nov 1;10(11):940–9. 

4. Wang Y, Kim R, Gunasekara DB, Reed MI, DiSalvo M, Nguyen DL, et al. Formation of Human Colonic Crypt Array by Application of Chemical Gradients Across a Shaped Epithelial Monolayer. Cell Mol Gastroenterol Hepatol. 2018;5(2):113–30. 

5. Zhong X-Y, Yu T, Zhong W, Li J-Y, Xia Z-S, Yuan Y-H, et al. Lgr5 positive stem cells sorted from small intestines of diabetic mice differentiate into higher proportion of absorptive cells and Paneth cells in vitro. Dev Growth Differ. 2015;57(6):453–65. 

6. Chandra L, Borcherding DC, Kingsbury D, Atherly T, Ambrosini YM, Bourgois-Mochel A, et al. Derivation of adult canine intestinal organoids for translational research in gastroenterology. BMC Biol. 2019 Apr 11;17(1):33. 

7. Cristina ML, Lehy T, Zeitoun P, Dufougeray F. Fine structural classification and comparative distribution of endocrine cells in normal human large intestine. Gastroenterology. 1978 Jul;75(1):20–8. 

8. Gunawardene AR, Corfe BM, Staton CA. Classification and functions of enteroendocrine cells of the lower gastrointestinal tract: Classification and functions of colorectal enteroendocrine cells. Int J Exp Pathol. 2011 Aug;92(4):219–31. 

9. Workman MJ, Gleeson JP, Troisi EJ, Estrada HQ, Kerns SJ, Hinojosa CD, et al. Enhanced Utilization of Induced Pluripotent Stem Cell-Derived Human Intestinal Organoids Using Microengineered Chips. Cell Mol Gastroenterol Hepatol. 2018;5(4):669-677.e2. 

10. Sáez de Guinoa J, Jimeno R, Gaya M, Kipling D, Garzón MJ, Dunn-Walters D, et al. CD1d-mediated lipid presentation by CD11c+ cells regulates intestinal homeostasis. EMBO J. 2018 Mar 1;37(5):e97537. 

11. Bullman S, Pedamallu CS, Sicinska E, Clancy TE, Zhang X, Cai D, et al. Analysis of Fusobacterium persistence and antibiotic response in colorectal cancer. Science. 2017 15;358(6369):1443–8. 

12. Hughes KR, Harnisch LC, Alcon-Giner C, Mitra S, Wright CJ, Ketskemety J, et al. Bifidobacterium breve reduces apoptotic epithelial cell shedding in an exopolysaccharide and MyD88-dependent manner. Open Biol. 7(1):160155. 

13. Yamakawa T, Tomita K, Sawai J. Characteristics of Biofilms Formed by Co-Culture of Listeria monocytogenes with Pseudomonas aeruginosa at Low Temperatures and Their Sensitivity to Antibacterial Substances. Biocontrol Sci. 2018;23(3):107–19. 

14. In J, Foulke-Abel J, Zachos NC, Hansen A-M, Kaper JB, Bernstein HD, et al. Enterohemorrhagic Escherichia coli Reduces Mucus and Intermicrovillar Bridges in Human Stem Cell-Derived Colonoids. Cell Mol Gastroenterol Hepatol. 2016 Jan 1;2(1):48-62.e3. 

15. Sontheimer-Phelps A, Chou DB, Tovaglieri A, Ferrante TC, Duckworth T, Fadel C, et al. Human Colon-on-a-Chip Enables Continuous In Vitro Analysis of Colon Mucus Layer Accumulation and Physiology. Cell Mol Gastroenterol Hepatol. 2019 Nov 26; 

16. Kasendra M, Luc R, Yin J, Manatakis DV, Apostolou A, Sunuwar L, et al. Organoid-derived Duodenum Intestine-Chip for preclinical drug assessment in a human relevant system. bioRxiv. 2019 Aug 5;723015. 

17. Kiela PR, Ghishan FK. Physiology of Intestinal Absorption and Secretion. Best Pract Res Clin Gastroenterol. 2016 Apr;30(2):145–59. 

18. Shin W, Kim HJ. Intestinal barrier dysfunction orchestrates the onset of inflammatory host–microbiome cross-talk in a human gut inflammation-on-a-chip. Proc Natl Acad Sci. 2018 Nov 6;115(45):E10539–47. 

19. Macierzanka A, Mackie AR, Krupa L. Permeability of the small intestinal mucus for physiologically relevant studies: Impact of mucus location and ex vivo treatment. Sci Rep. 2019 Nov 26;9(1):1–12.

---

## [Decision Letter · Decision Letter 1]

24 Mar 2020

Recapitulation of the Accessible Interface of Biopsy-Derived Canine Intestinal Organoids to Study Epithelial-Luminal Interactions

PONE-D-20-00243R1

Dear Dr. Kim,

We are pleased to inform you that your manuscript has been judged scientifically suitable for publication and will be formally accepted for publication once it complies with all outstanding technical requirements.

With kind regards,

Mária A. Deli, M.D., Ph.D.

Academic Editor

PLOS ONE

Additional Editor Comments (optional):

Reviewers' comments:

Reviewer's Responses to Questions

**Comments to the Author**

1. If the authors have adequately addressed your comments raised in a previous round of review and you feel that this manuscript is now acceptable for publication, you may indicate that here to bypass the “Comments to the Author” section, enter your conflict of interest statement in the “Confidential to Editor” section, and submit your "Accept" recommendation.

Reviewer #1: All comments have been addressed

Reviewer #2: All comments have been addressed

2. Is the manuscript technically sound, and do the data support the conclusions?

Reviewer #1: Yes

Reviewer #2: Yes

3. Has the statistical analysis been performed appropriately and rigorously? 

Reviewer #1: Yes

Reviewer #2: Yes

4. Have the authors made all data underlying the findings in their manuscript fully available?

Reviewer #1: Yes

Reviewer #2: Yes

5. Is the manuscript presented in an intelligible fashion and written in standard English?

Reviewer #1: Yes

Reviewer #2: Yes

6. Review Comments to the Author

Reviewer #1: (No Response)

Reviewer #2: The authors have addressed all my concerns really well. Novel experiments and proper explanation/interpretation of results.

I have one minor comment: graphs on figure 6C and D should not be present with errors since they were generated from 2 bilogical replicates.

7. PLOS authors have the option to publish the peer review history of their article (what does this mean?). If published, this will include your full peer review and any attached files.

Reviewer #1: No

Reviewer #2: No

---

## [Editor Report · Acceptance letter]

6 Apr 2020

PONE-D-20-00243R1 

Recapitulation of the Accessible Interface of Biopsy-Derived Canine Intestinal Organoids to Study Epithelial-Luminal Interactions 

Dear Dr. Kim:

I am pleased to inform you that your manuscript has been deemed suitable for publication in PLOS ONE. Congratulations! Your manuscript is now with our production department. 

With kind regards,

on behalf of

Dr. Mária A. Deli 

Academic Editor

PLOS ONE